# Design and Synthesis of a Novel 4-aryl-N-(2-alkoxythieno [2,3-*b*]pyrazine-3-yl)-4-arylpiperazine-1-carboxamide DGG200064 Showed Therapeutic Effect on Colon Cancer through G2/M Arrest

**DOI:** 10.3390/ph15050502

**Published:** 2022-04-20

**Authors:** Eun-Sil Lee, Nayeon Kim, Joon Hee Kang, Aizhan Abdildinova, Seon-Hyeong Lee, Myung Hwi Lee, Nam Sook Kang, Tae-Sung Koo, Soo-Youl Kim, Young-Dae Gong

**Affiliations:** 1Innovative Drug-Like Library Research Center, Dongguk University, Pil-dong 3-ga, Jung-gu, Seoul 100-715, Korea; yantil@hanmail.net (E.-S.L.); nykim0117@gmail.com (N.K.); aizhik.a91@gmail.com (A.A.); 2Division of Cancer Biology, National Cancer Center, Research Institute, Goyang 410-769, Korea; wnsl2820@gmail.com (J.H.K.); shlee1987@gmail.com (S.-H.L.); 3Graduate School of New Drug Discovery and Development, Chungnam National University, Daehak-ro 99, Yuseong-gu, Daejeon 305-764, Korea; sym1019@hanmail.net (M.H.L.); nskang@cnu.ac.kr (N.S.K.); kootae@cnu.ac.kr (T.-S.K.)

**Keywords:** lead compound, cell cycle arrest, CDC4-cJUN, G2/M arrest, anticancer activity, colorectal cancer

## Abstract

Cancer cells are characterized by an abnormal cell cycle. Therefore, the cell cycle has been a potential target for cancer therapeutic agents. We developed a new lead compound, **DGG200064** (**7c**) with a 2-alkoxythieno [2,3-*b*]pyrazine-3-yl)-4-arylpiperazine-1-carboxamide core skeleton. To evaluate its properties, compound **DGG200064** was tested in vivo through a xenograft mouse model of colorectal cancer using HCT116 cells. The in vivo results showed high cell growth inhibition efficacy. Our results confirmed that the newly synthesized **DGG200064** inhibits the growth of colorectal cancer cells by inducing G2/M arrest. Unlike the known cell cycle inhibitors, **DGG200064** (GI_50_ = 12 nM in an HCT116 cell-based assay) induced G2/M arrest by selectively inhibiting the interaction of FBXW7 and c-Jun proteins. Additionally, the physicochemical properties of the lead compounds were analyzed. Based on the results of the study, we suggested further development of **DGG200064** as a novel oral anti-colorectal cancer drug.

## 1. Introduction

The cell cycle is a sequence of events in which the cell grows and replicates, and it is controlled by a complex network of interactions between various factors. The cell cycle is composed of four stages: G0/G1, S, G2, and M. Activities in each stage are regulated by cyclins and cyclin-dependent kinases (CDKs) at key checkpoints. The activity of CDKs is induced by cell growth signals and can be inhibited by cell responses to DNA damage. Unlike in the normal cells, the cell cycle in cancer cells is unregulated as a result of mutations and genetic deficiencies of cancer cells [1]. Unregulated CDKs activities have been reported in a majority of cancers [2]. Therefore, CDKs and the cell cycle have been a potential target for cancer therapeutic agents over the years [3,4]. Anticancer drugs that target the cell cycle are classified as pan-CDK inhibitors, CDK4/6 selective inhibitors, CHK1 inhibitors, WEE1 inhibitors, PLK inhibitors, and Aurora inhibitors. Pan-CDK inhibitors inhibit all stages of the cell cycle, and they include Dinaciclib [5,6], R-roscovitine [7], and Flavopiridol [8]. CDK4/6 selective inhibitors induce G1 arrest and inhibit the cancer cell growth, and they include Palbociclib and Ribociclib [9,10,11]. CHK1 inhibitor LY2606368 induces DNA double-strand breaks in the S phase and causes mitotic cell death [12]. The combination of WEE1 inhibitors, such as AZD1775 with a DNA damaging agent, has been reported to induce apoptosis [13,14]. Rigosertib, a PLK inhibitor, induces apoptosis by causing spindle abnormality and mitotic arrest [15,16]. Volasertib causes G2/M arrest and cell death [15,16]. Aurora inhibitors, such as Alisertib, causes spindle abnormality and mitotic arrest [17,18].

Efforts to modulate cell-cycle arrest in G2/M are still in progress. Studies show that G2/M arrest defects may allow a damaged cell to enter mitosis and undergo apoptosis, which can lead to increased cytotoxicity; while efforts to induce G2/M arrest have been correlated with enhanced apoptosis [19]. Tyagi et al. showed Silibinin in combination with Doxorubicin can serve as a G2/M cell-cycle checkpoint regulator [20]. Cabrera et al. reported on benzophenone thiosemicarbazone derivative T44Bf induced G2/M cell cycle arrest and selectively induced apoptosis of acute leukemia cells [21]. Chen et al. reported on luteolin as a modulator of G2/M cell cycle arrest and apoptosis in human colon cancer cells and xenografts [22]. In addition, Lee et al. reported on Ganetespib inducing G2/M cell cycle arrest and apoptosis in gastric cancer cells [23].

To identify small molecule inhibitors of the G2/M-specific cell cycle, we screened 240 drug-like compounds via cell-based anti-colorectal activity HCT116 reporter assay. The compounds were selected from our in-house library of 30,000 structurally diverse druggable heterocyclic compounds including benzopyrans [24], oxazoles [25], pyrazoles [26], oxadiazoles [27,28], thiadiazoles [29,30,31], various thiazoles [32,33,34,35,36], pyrimidinediones [37], and benzimidazoles [38,39]. Inhibition of cell proliferation of colorectal cancer cell HCT116 was screened at a compound concentration of 5 μM; compounds that reproducibly inhibited growth by over 100% were selected. In the first round of screening, 2-alkoxythieno[2,3-*b*]pyrazine-3-yl)-4-arylpiperazine-1-carboxamide derivatives showed good inhibitory activity with IC_50_ below 1 μM. Therefore, to optimize the structural features of compounds after primary screening results, we designed several target compounds for the 2-alkoxythieno[2,3-*b*]pyrazine-3-yl)-4-arylpiperazine-1-carboxamide library.

In this study, we developed and evaluated a variety of 4-arylpiperazine-1-carboxamide based compounds as potential inhibitors of colorectal cancer with **DGG200064** showing the best anti-cancer effect. We also demonstrated that this compound induces G2/M arrest by selectively inhibiting c-Jun and FBXW7 interaction (Figure 1), rather than the modulation of CDK activity, as it was previously reported [2]. Herein, we present the synthesis and biological evaluation of the 4-arylpiperazine-1-carboxamides as a potential treatment of colon cancer as shown in the G2/M cell cycle arrest research concept diagram.

## 2. Results and Discussion

The synthesis scheme of the key intermediates, 6-chloro-2-methoxythieno[2,3-*b*]pyrazin-3-amine derivatives **5a**–**5d**, is shown in Figure 1. Compounds **5a**–**5d** were synthesized from the corresponding 2,6-dichlorothieno[2,3-*b*]pyrazin-3-amine **4** according to the procedure reported [40]. First, the Bromo group of the 5-bromo-6-chloropyrazin-2-amine **1** was subjected to the trimethylsilyl ethynylation with trimethylsilyl acetylene in the presence of PdCl_2_(dppf)_2_, CuI, and NEt_3_ in THF, producing 6-chloro-5-((trimethylsilyl)ethynyl)pyrazin-2-amine **2**. The cyclization of 6-chloro-5-((trimethylsilyl)ethynyl)pyrazin-2-amine **2** via Na_2_S∙5H_2_O produced thieno[2,3-*b*]pyrazin-3-amine **3**. The bicyclic compound **3** underwent chlorination reaction with *N*-chlorosuccinimide (NCS) to provide 2,6-dichlorothieno[2,3-*b*]pyrazin-3-amine **4** in good yield. The key intermediates **5a**–**5d** were prepared through two pathways shown in Figure 1. First, key intermediates **5a**–**5b**, 6-chloro-2-alkoxythieno[2,3-*b*]pyrazin-3-amine derivatives, were synthesized by alkoxylation reaction of **4** with NaOMe in MeOH (for **5a**) or with NaOEt in EtOH (for **5b**). Further, the dichlorination of the compounds **5a**–**5b** at 6-chloro position in the presence of Pd/C and NH_4_CO_2_H in EtOH under microwave irradiation (MW) produced 6-hydrogenated intermediates, 2-alkoxythieno[2,3-*b*]pyrazin-3-amine derivatives **5c**–**5d**.

Following substitution reactions of 2-methoxythieno[2,3-*b*]pyrazin-3-amines **5**, which were proceeded by the phenyl chloroformate treatment in the presence of pyridine at room temperature (rt), produced phenyl(2-alkoxythieno[2,3-*b*]pyrazin-3-yl)carbamates **6** (Figure 2). Then, we introduced various phenyl piperazine derivatives into the carbamate position of the intermediate **6** in the presence of NEt_3_ in CH_3_CN to obtain our target compounds, 4-phenyl-*N*-(thieno[2,3-*b*]pyrazin-3-yl)piperazine-1-carboxamides **7a**–**7i**, in good to high yields as shown in Table 1.

### 2.1. Cell-Based SRB Assay: Cytotoxicity and SAR Analysis

A series of 4-phenyl-*N*-(thieno[2,3-*b*]pyrazin-3-yl)piperazine-1-carboxamides derivatives **7a**–**7i** were tested for the colon cancer cell growth inhibition effect using the SRB assay (Table 2). The GI_50_ values of the selected compounds were lower than 1 μM and varied significantly depending on the substitution groups at thieno[2,3-*b*]pyrazine and phenyl ring. Compounds **7a** and **7b**, containing chlorine at the C6 position of thieno[2,3-*b*]pyrazine ring, have weaker anticancer activity than its proton-containing derivatives **7c**–**7i**. On the other hand, **7c** has shown higher inhibition activity than **7d** with methyl substituents at the C3 and C5 positions of the phenyl ring. The introduction of two fluorine atoms drastically decreased the efficacy of compound **7e**. However, the substitution of the methoxy group to the methyl group at the C5 position of the phenyl ring significantly improved the anticancer activity (compound **7f**). Substitutions with one fluorine atom in the phenyl ring could not improve the inhibition effect of the compounds **7g**–**7h**. Similarly, switching the methoxy group with ethoxy at the C2 position of the thieno[2,3-*b*]pyrazine ring, compound **7i**, also did not improve efficacy probably due to the steric effect. Additionally, we had introduced various bulky alkyls and trifluoromethyl groups to C2 position of the thieno[2,3-*b*]pyrazine ring dramatically decreased efficacy.

Among the nine final compounds, **7c** and **7f** showed the best inhibition activity, as shown in Table 2, and the drug action mechanism was confirmed via **7c**. PK data of the compounds **7c** and **7f** showed relatively low bioavailability (20.11% F for **7c**, 21.83% F for **7f**) as well as low C_max_ (0.076 for **7c**, 0.195 for **7f**). A detailed analysis of the PK optimization process is in the PK results section.

### 2.2. Induced G2/M Cell Cycle Arrest by DGG200064

Many anticancer drugs induce apoptosis through programmed signaling in tumor cells. To test whether the newly developed 4-phenyl-*N*-(thieno[2,3-*b*]pyrazin-3-yl)piperazine-1-carboxamide derivatives could induce apoptosis, we conducted a FACS analysis of HCT116 cells with the representative compound **7c** or **DGG200064**. Cell cycle distribution of HCT116 cells was investigated with different concentrations of **DGG200064** for 6 h. Compound **DGG200064** did not cause apoptosis, but it induced G/2M arrest in a dose-dependent manner in HCT116 cells (Figure 2a). Additionally, **DGG200064** was tested for induced G2/M arrest in the other types of colon cancer cells in concentrations of 50 nM for 6 h (Figure 2b).

### 2.3. c-Jun Stabilization by DGG200064

HCT116 and DLD1 cells were exposed to increasing concentrations of **DGG200064** (0, 10, 50, 100 nM) for 6 h. Immunoblotting showed a dose-dependent increase in cyclin B1, c-Jun, and p-c-Jun by **DGG200064** (Figure 3a). Moreover, there were increased levels of c-Jun by **DGG200064** (50 nM) treatment in other 6-colon cancer cells (Appendix A). c-Jun is a component of the transcription factor activator protein 1 (AP-1), which is activated by a variety of extracellular stimuli, such as growth factors and UV irradiation. In cancer cells, activation of c-Jun promotes down-stream target gene transcription, and it is involved in cell proliferation, growth, division, and apoptosis [41]. Immunocytochemical staining of c-Jun in the presence of **DGG200064** (50 nM) showed increased levels of c-Jun level in HCT116 cells (Figure 3b). To test whether **DGG200064** could induce G2/M cell cycle arrest through the increasing of c-Jun level, the two genes were silenced using c-Jun siRNAs and treated with **DGG200064** in HCT116 and DLD1 cells (Figure 3c). When c-Jun was silenced, the amount of cyclin B protein treated by **DGG200064** decreased. This result suggests that G2/M cell cycle arrest is induced by c-Jun, which is increased by the effect of **DGG200064**. Next, we investigated if the c-Jun stabilization was induced through inhibition of protein degradation by **DGG200064.** Combined treatment with both siRNA for FBXW7 (E3 ligase of c-Jun) and **DGG200064** resulted in changes for both c-Jun and cyclin B in HCT116 and DLD1 cells (Figure 3d). c-Jun and cyclin B1 were not observed after treatment with **DGG200064**. This finding suggested that **DGG200064** inhibited the degradation of c-Jun by the E3 ligase, FBXW7.

### 2.4. Selective Inhibition of c-Jun Ubiquitination through Interruption of FBXW7/c-Jun Interaction

To investigate whether the ubiquitination of c-Jun is suppressed through the inhibition of c-Jun and FBXW7 binding by **DGG200064**, c-Jun immunoprecipitation was done to observe protein–protein interaction (Figure 4). E3 ligase, such as FBXW7, binds the protein to the ubiquitin chain through covalent bonds and rapidly degrades the protein through 26S proteasome [42]. When treated with MG132 as a proteasome inhibitor, the ubiquitination of c-Jun increased compared to both control HCT116 and DLD1 cells. In contrast, the c-Jun ubiquitination was inhibited by **DGG200064**. The binding between c-Jun and FBXW7 was reduced with **DGG200064** treatment. The FBXW7 protein, also known as CDC4, is a component of the SCF complex ubiquitin ligase and is known as a modulator of several substrates including cyclin E, c-Myc, c-Jun, and Notch [43]. To check the selectivity of **DGG200064** towards FBXW7 substrates, the change of c-Jun, cyclin E, and c-Myc by **DGG200064** was analyzed. After treatment with **DGG200064** (0, 10, 50, 100 nM) for 6 h in HCT116 cells, immunoblotting showed an increased level of c-Jun but the cyclin E and c-Myc levels were not affected (Appendix A). Cyclin E immunoprecipitation was done to test whether the ubiquitination of cyclin E by FBXW7 is inhibited by **DGG200064** in HCT116 (Appendix A). There were no changes in cyclin E ubiquitination with **DGG200064** treatment. **DGG200064** was found to selectively inhibit the ubiquitination of c-Jun by interrupting the binding between FBXW7 and c-Jun.

### 2.5. Identification of the Interaction Inhibition Site between FBXW7 and c-Jun by Docking Study

The stabilization of c-Jun was induced by the suppression of c-Jun ubiquitination, through inhibition of binding between FBXW7 and c-Jun by **DGG200064**. To identify the binding site on FBXW7 and c-Jun, we conducted a docking study. The crystal structure of human FBXW7 bound to SKP1 and Cyclin E (PDB ID:2OVQ) complex is known [44]. The c-Jun, **7a**, **7d**, **7f**, and **DGG200064 (7c)** were docked at the binding site of FBXW7 (Figure 5a). In the docked mode, an aliphatic side chain of glutamic acid at the *p*-1 position of c-Jun occupied the FBXW7 hydrophobic pocket while the carboxylate group was located adjacent to the solvent-exposed side chain of R689 on the FBXW7 surface. This was in agreement with a previous study [44]. R689 was involved in electrostatic and hydrogen bond interactions whereas A599 and A626 showed hydrophobic interactions with c-Jun.

Docking results showed that highly potent compounds, **7f** (GI_50_ = 0.0019 μM) and **DGG200064** (GI_50_ = 0.012 μM), interacted with T628 in addition to R689. Compounds with low potency such as **7a** (GI_50_ ≥ 0.1 μM) and **7e** (GI_50_ ≥ 0.1 μM) interacted with R689 but could not interact with T628; this might be the reason for their low potency. For the most potent compound **7f**, one of the N atoms of thienopyrazine moiety displayed hydrogen bonding with side chain NH group of R689 at a distance of 2.06 Å. NH group of amide linkage exhibited hydrogen bond with side chain CO group of D642 at a distance of 1.54 Å. Piperazine ring demonstrated alkyl interaction with A626. Benzene ring showed pi-sigma interaction with T628. Methoxy and methyl groups of the benzene ring displayed alkyl interactions with L583 and W673. The compound **DGG200064** also showed interactions similar to that of **7f**. Unlike **7f**, **DGG200064** possesses a methoxy group instead of the methyl group at the benzene ring. This replacement affected the alkyl interaction of the other methoxy group with L583. The distance between the methoxy group and L583 was found to be longer for **DGG200064** (5.48 Å) than **7f** (4.91 Å). The longer distance could be responsible for the slightly lower activity of **DGG200064** as compared to **7f**.

In comparison to high potent compounds (**7f** and **DGG200064**), relatively low potency compounds, **7a** and **7b**, have a chlorine atom instead of a hydrogen atom at the thienopyrazine moiety as well as **7i** have an ethoxy group instead of a methoxy group at the thienopyrazine moiety. Low potency compound **7e** also have di-fluorine groups instead of a di-methoxy group at the phenyl piperazine moiety. This position is surrounded by S625 and A626. Although they maintained hydrogen bond interactions with R689 and D642, the shift leads to the loss of hydrophobic interactions with T628, which might be the reason for their lower potency.

Through the docking study, the key amino acid residues in the binding site of c-Jun and FBXW7 were identified as Arg689 and Thr628. To evaluate the possibility of a double mutant form of FBXW7 (R689G & T628A), transiently expressed in HCT116 cells, we confirmed the binding of c-Jun using anti-Myc-tag antibody immunoprecipitation (Figure 5b). The mutant FBXW7 decreased binding c-Jun as well as the **DGG200064** effects.

### 2.6. DGG200064 Treatment Abrogated CRC in Xenograft Models with an Increase in the c-Jun Level

We tested the inhibition effect by **DGG200064** in the HCT116 mouse tumor models (Figure 6a). Cultured HCT116 cells were injected subcutaneously near the scapulae of 6-week-old female nude BALB/c mice. Oral administration of **DGG200064** (60, 120 mg/kg) began when tumors reached a volume of 100 mm^3^ and proceeded for 6 days/week. After administering for 21 days, it showed a significant difference in the control-vehicle group and the **DGG200064** treatment groups. Mice were then injected intraperitoneally with BrdU and anesthetized 2 h later. They were then perfused with PBS and killed. To identify the level of c-Jun by **DGG200064**, HCT116 tumors harvested from mice were fixed and analyzed by immunohistochemical staining for Hematoxylin and c-Jun. This led c-Jun expression to increase 1.2-and 2-fold in the 60 mg/kg and 120 mg/kg groups, respectively, compared to the control group (Figure 6b).

### 2.7. Pharmacokinetic Analysis and Study

The pharmacokinetic properties (PK) of compounds **7c** with the highest efficacy was checked in SD rats and ICR Mice as shown in Table 3. The selected compounds showed similar bioavailability compared to the commercially available general oral drugs. After testing, the compounds showed slight differences in some pharmacokinetic parameters including the maximal plasma concentrations (C_max_), the half-lives (T_1/2_), the systemic clearance (CL), the area under the curves (AUC), and the oral bioavailability (F). Especially bioavailability with (F) showed 20.11% and 13.8% underwent PK in SD rats and ICR Mice. These kind of PK data were shown lower data than our expectations.

### 2.8. Evaluation of Physicochemical Properties

The physicochemical properties of the lead compounds **DGG200064** (**7c**) was checked (Table 4). As it was stated by Lipinski, molecular properties are closely related to the oral bioavailability of a drug [45]. Especially, water solubility and membrane permeation are considered as a basic requirement for oral bioavailability. Membrane permeability is known to have a good correlation with CLogP [46]. Hann et al. suggested lead-likeness of the compound with CLogP values ≤4.2, and solubility ranging from −5 to 0.5 [47]. Through experiments, actual values including pKa, logP, permeability, and solubility were obtained. The physicochemical properties of both compounds are very similar and showed good potential as oral drugs. Especially, high permeability and moderate solubility correspond on lead-likeness [47] of the compounds (Figure 7). Additionally, the lead compounds showed stable values for CYP and hERG inhibition showed druggable properties at 10 uM towards different isozymes (Figure 8).

## 3. Materials and Methods

This study was reviewed and approved by the Institutional Animal Care and Use Committee (IACUC) of the National Cancer Center Research Institute.

### 3.1. General Methods

Commercially available reagents and solvents were used without further processing. Thin-layer chromatography analysis was used to monitor reactions using thin-layer plates on silica gel 60 F_254_ (Merck KGaA, Darmstadt, Germany). Flash column chromatography was performed using silica gel 60 (230–400 mesh). ^1^H NMR and ^13^C NMR spectra were recorded in δ units relative to the deuterated solvent as an internal reference using a 500 MHz NMR instrument (Bruker, Billerica, MA, USA). Liquid chromatography-tandem mass spectrometry (Agilent 6460 Triple Quad LC/MS, Santa Clara, CA, USA) analysis was performed using electrospray ionization (ESI) mass spectrometer with photodiode array detector (PDA). High-resolution mass spectrometry spectra were obtained using a TOF LC/MS system (Agilent 6550 iFunnel Q-TOF LC/MS).

### 3.2. Synthesis of 6-chloro-5-((trimethylsilyl)ethynyl)pyrazin-2-amine ***2***

To a solution of 5-bromo-6-chloropyrazin-2-amine **1** (4.2 g, 20 mmol, 1 eq) in THF (100 mL), PdCl_2_(dppf)_2_ (1.3 g, 1.6 mmol, 0.08 eq) and CuI (0.19 g, 1.0 mmol, 0.05 eq) were added under nitrogen atmosphere. Then, trimethylsilylacetylene (3.6 mL, 26 mmol, 1.3 eq) was added, followed by NEt_3_ (16.7 mL, 120 mmol, 6 eq). The mixture was stirred for 2 h at 80 °C. After completion, the reaction mixture was diluted with water and extracted with EA (40 mL × 3), and the organic layers were combined, washed with brine, dried over Na_2_SO_4_, filtered, and concentrated. The residue was purified with column chromatography (Hex/EA = 5:1) to yield the desired compound **2** (3.16 g, 70%). ^1^H NMR (500 MHz, CDCl_3_) δ 7.84 (s, 1H), 4.85 (s, 2H), 0.28 (s, 9H). ^13^C NMR (126 MHz, CDCl_3_) δ 152.38, 149.08, 129.97, 126.87, 100.37, 99.68, 0.30. HRMS [ESI^+^]: calcd for C_9_H_12_ClN_3_Si, 226.0562 [M + H]^+^; found, 226.0562.

### 3.3. Synthesis of thieno [2,3-b]pyrazin-3-amine ***3***

To a solution of 6-chloro-5-((trimethylsilyl)ethynyl)pyrazin-2-amine **2** (5.0 g, 22.2 mmol, 1 eq) in DMF (50 mL), Na_2_S∙5H_2_O (14.9 g, 88.8 mmol, 4 eq) was added under nitrogen and stirred for 2 h at 90 °C. After the reaction was complete, the mixture was diluted with water and extracted with EA (50 mL × 5), and the organic layers were combined, washed with brine, dried over Na_2_SO_4_, filtered, and concentrated. The residue was purified with column chromatography (Hex/EA = 1:1) to yield compound **3** (2.3 g, 69%). ^1^H NMR (500 MHz, CDCl_3_) δ 8.07 (s, 1H), 7.36 (d, *J* = 5.5 Hz, 1H), 7.32 (d, *J* = 5.5 Hz, 1H), 4.70 (s, 2H). ^13^C NMR (126 MHz, CDCl_3_) δ 153.86, 151.15, 141.53, 131.15, 123.81, 122.54. HRMS [ESI^+^]: calcd for C_6_H_5_N_3_S, 152.0277 [M + H]^+^; found, 152.0278.

### 3.4. Synthesis of 2,6-dichlorothieno [2,3-b]pyrazin-3-amine ***4***

To a solution of thieno[2,3-*b*]pyrazin-3-amine **3** (0.61 g, 4.0 mmol, 1 eq) in ACN (60 mL), NCS (1.17 g, 8.8 mmol, 2.2 eq) was added under nitrogen and stirred for 30 min at 70 °C. After the reaction was complete, the mixture was diluted with water and extracted with EA (30 mL × 3), and the organic layers were combined, washed with brine, dried over Na_2_SO_4_, filtered, and concentrated. The residue was triturated with Et_2_O to yield compound **4** (0.61 g, 70%). Compound 4 was used to synthesis **5a–d** without further purification. ^1^H NMR (500 MHz, CDCl_3_) δ 7.13 (s, 1H), 5.10 (s, 2H). ^13^C NMR (126 MHz, DMSO-d_6_) δ 151.95, 150.20, 137.56, 133.32, 127.37, 121.61 HRMS [ESI^+^]: calcd for C_6_H_3_Cl_2_N_3_S, 219.9497 [M + H]^+^; found, 219.9496.

### 3.5. Synthesis of the Key Intermediates, 6-chloro-2-alkoxyoxythieno [2,3-b]pyrazin-3-amines ***5a***, ***5b***

To a solution of 2,6-dichlorothieno[2,3-*b*]pyrazin-3-amine **4** (0.82 g, 3.8 mmol, 1.0 eq) in MeOH (15 mL), NaOMe (7.06 mL, 38 mmol, 10 eq, 30 wt. % in MeOH) was added and stirred for 1 h at 90 °C. The MeOH was evaporated under reduced pressure, then the mixture was diluted with water and extracted with EA (20 mL × 3). The organic layers were combined, washed with brine, dried over Na_2_SO_4_, filtered, and concentrated. The residue was purified with column chromatography (Hex/EA = 5:1) to yield compound **5a** (0.49 g, 60%). Following the procedure described for **5a**, compound **4** with NaOEt (14.16 mL, 38 mmol, 10 eq, 21 wt. % in EtOH) provided compound **5b** (0.61 g, 70%).

6-Chloro-2-methoxythieno[2,3-*b*]pyrazin-3-amine **5a**: ^1^H NMR (500 MHz, CDCl_3_) δ 7.05 (s, 1H), 4.93 (s, 2H), 4.04 (s, 3H). ^13^C NMR (126 MHz, CDCl_3_) δ 148.06, 144.02, 143.22, 136.34, 127.53, 120.89, 53.95. HRMS [ESI^+^]: calcd for C_7_H_6_ClN_3_OS, 215.9993 [M + H]^+^; found, 215.9993.

6-Chloro-2-ethoxythieno[2,3-*b*]pyrazin-3-amine **5b**: ^1^H NMR (500 MHz, CDCl_3_) δ 7.03 (s, 1H), 4.93 (d, *J* = 44.2 Hz, 2H), 4.46 (q, *J* = 7.1 Hz, 2H), 1.45 (*t*, *J* = 7.1 Hz, 3H). ^13^C NMR (126 MHz, CDCl_3_) δ 146.65, 142.82, 142.22, 135.29, 126.32, 119.89, 61.63, 13.42. HRMS [ESI^+^]: calcd for C_8_H_8_ClN_3_OS, 230.0149 [M + H]^+^; found, 230.0150.

### 3.6. Synthesis of 2-alkoxyoxythieno [2,3-b]pyrazin-3-amines ***5c***, ***5d***

To a solution of **5a** (0.43 g, 2.0 mmol, 1 eq) in EtOH (15 mL), 10% Pd/C (0.85 g), NH_4_CO_2_H (1.51 g, 24 mmol, 12 eq) were added. The mixture was irradiated in a microwave for 30 min at 100 °C. After cooling, the mixture was evaporated under reduced pressure, filtered over Celite, and concentrated. The residue was purified with column chromatography (Hex/EA = 5:1) to yield compound **5c** (0.32 g, 88%). Following the procedure described for **5d**, the compound **5b** (0.46 g, 2.0 mmol) provided the desired key intermediate **5d** (0.27 g, 70%) in the same reaction condition.

2-Methoxythieno [2,3-*b*]pyrazin-3-amine **5c**: ^1^H NMR (500 MHz, CDCl_3_) δ 7.24 (d, *J* = 5.9 Hz, 1H), 7.19 (d, *J* = 5.9 Hz, 1H), 5.01 (s, 2H), 4.06 (s, 3H). ^13^C NMR (126 MHz, CDCl_3_) δ 147.84, 145.19, 143.12, 137.37, 121.84, 121.74, 53.81. HRMS [ESI^+^]: calcd for C_7_H_7_N_3_OS, 182.0383 [M + H]^+^; found, 182.0384.

2-Ethoxythieno [2,3-*b*]pyrazin-3-amine **5d**: ^1^H NMR (500 MHz, CDCl_3_) δ 7.23 (d, *J* = 5.9 Hz, 1H), 7.17 (d, *J* = 5.9 Hz, 1H), 5.02 (s, 2H), 4.49 (q, *J* = 7.1 Hz, 2H), 1.46 (*t*, *J* = 7.1 Hz, 3H). ^13^C NMR (126 MHz, CDCl_3_) δ 147.46, 144.92, 143.15, 137.34, 121.76, 121.66, 62.46, 14.49. HRMS [ESI^+^]: calcd for C_8_H_9_N_3_OS, 196.0539 [M + H]^+^; found, 196.0540.

### 3.7. Synthesis of diphenyl(6-substituted-2-alkoxythieno [3,2-b]pyrazin-3-yl)imino-dicarbonates ***6a–6c***

To a solution of **5a** (0.48 g, 2.23 mmol, 1 eq) in CH_2_Cl_2_ (30 mL), pyridine (2.52 mL, 31.22 mmol, 14 eq) and phenyl chloroformate (1.26 mL, 10.04 mmol, 4.5 eq) were added at 0 °C. The reaction mixture was stirred for 1 h at room temperature. After the reaction was complete, the mixture was concentrated under reduced pressure. The residue was purified with column chromatography (Hex/EA = 7:1) to yield compound **6a** (0.86 g, 85%). Following the procedure described for **6a**, compound **5c** (0.40 g, 2.23 mmol) provided compound **6b** (0.69 g, 74%).

**Diphenyl(6-chloro-2-methoxythieno[2,3-*b*]pyrazin-3-yl)imino-dicarbonate 6a**: ^1^H NMR (500 MHz, CDCl_3_) δ 7.36 (dt, *J* = 10.7, 2.1 Hz, 4H), 7.28 (s, 1H), 7.23 (dd, *J* = 10.5, 4.0 Hz, 2H), 7.16–7.12 (m, 4H), 4.16 (s, 3H). ^13^C NMR (126 MHz, CDCl_3_) δ 154.94, 150.23, 149.81, 146.70, 143.84, 139.44, 133.21, 129.50, 126.45, 121.24, 120.93, 54.76. HRMS) [ESI^+^]: calcd for C_21_H_14_ClN_3_O_5_S, 456.0415 [M + H]^+^; found, 456.0415.

Diphenyl(2-methoxythieno[3,2-*b*]pyrazin-3-yl)imino-dicarbonate **6b**: ^1^H NMR (500 MHz, CDCl_3_) δ 7.88 (d, *J* = 6.0 Hz, 1H), 7.41 (d, *J* = 6.0 Hz, 1H), 7.36 (*t*, *J* = 7.9 Hz, 4H), 7.24 (dd, *J* = 10.9, 4.0 Hz, 2H), 7.15 (d, *J* = 8.4 Hz, 4H), 4.19 (s, 3H). ^13^C NMR (126 MHz, CDCl_3_) δ 154.41, 150.27, 149.93, 146.96, 144.97, 133.25, 132.99, 129.48, 126.41, 121.67, 121.30, 54.60. HRMS [ESI^+^]: calcd for C_21_H_15_N_3_O_5_S, 422.0805 [M + H]^+^; found, 422.0805.

Following the procedure described for **6a**, compound **5d** provided compound **6c** (0.46 g, 65%).

Phenyl (2-ethoxythieno [2,3-*b*]pyrazin-3-yl)carbamate **6c**: ^1^H NMR (500 MHz, CDCl_3_) δ 7.89 (s, 1H), 7.57 (d, *J* = 6.0 Hz, 1H), 7.43–7.37 (m, 2H), 7.28–7.25 (m, 4H), 4.58 (q, *J* = 7.1 Hz, 2H), 1.52 (*t*, *J* = 7.1 Hz, 3H). ^13^C NMR (126 MHz, CDCl_3_) δ 150.40, 149.56, 148.06, 144.73, 141.20, 134.38, 129.40, 127.85, 125.89, 121.54, 121.33, 63.33, 14.50. HRMS [ESI^+^]: calcd for C_15_H_13_N_3_O_3_S, 316.075 [M + H]^+^; found, 316.075.

### 3.8. Synthesis of the Final Products, N-(6-substituted-2-alkoxythieno [2,3-b]pyrazin-3-yl)-4-(3,5-dimethoxyphenyl)piperazine-1-carboxamide ***7a–7i***

To a solution of **6a** (0.45 g, 1.0 mmol, 1 eq) in ACN (20 mL), NEt_3_ (0.42 mL, 3.0 mmol, 3 eq) and R_3_-Ph-piperazine (0.67 g, 3.0 mmol, 3 eq) were added. The reaction mixture was stirred for 1 h at 60 °C. After the reaction was complete, the whole mixture was diluted with water and extracted with EA (20 mL × 3), and the organic layers were combined, washed with brine, dried over Na_2_SO_4_, filtered, and concentrated. The residue was purified with column chromatography (Hex/EA = 1:1) to yield compound **7a** (0.31 g, 66%). Following the procedure described for **7a**, **6a**, **6b**, and **6c** provided compounds **7b** (0.27 g, 63%), **7c** (0.33 g, 78%), **7d** (0.34 g, 87%), **7e** (0.38 g, 95%), **7f** (0.33 g, 79%), **7g** (0.38 g, 90%), **7h** (0.38 g, 95%), **7i** (0.35 g, 78%).

*N*-(6-chloro-2-methoxythieno[2,3-*b*]pyrazin-3-yl)-4-(3,5dimethoxyphenyl)piperazine-1-carboxamide **7a**: ^1^H NMR (500 MHz, CDCl_3_) δ 7.13 (d, *J* = 4.1 Hz, 2H), 6.10 (d, *J* = 1.8 Hz, 2H), 6.07 (s, 1H), 4.08 (s, 3H), 3.79 (s, 6H), 3.75–3.69 (m, 4H), 3.29–3.23 (m, 4H). ^13^C NMR (126 MHz, CDCl_3_) δ 161.58, 153.05, 152.84, 148.81, 143.90, 139.48, 136.66, 132.41, 120.65, 95.59, 92.23, 55.31, 54.44, 49.19, 44.63. HRMS [ESI^+^]: calcd for C_20_H_22_ClN_5_O_4_S, 464.1154 [M + H]^+^; found, 464.1154.

*N*-(6-chloro-2-methoxythieno[2,3-*b*]pyrazin-3-yl)-4-(3,5-dimethylphenyl)piperazine-1-carboxamide **7b**: ^1^H NMR (500 MHz, CDCl_3_) δ 7.12 (s, 2H), 6.58 (s, 3H), 4.08 (s, 3H), 3.77–3.66 (m, 4H), 3.27–3.20 (m, 4H), 2.29 (s, 6H). ^13^C NMR (126 MHz, CDCl_3_) δ 153.03, 151.06, 148.82, 143.94, 139.46, 138.90, 136.70, 132.39, 122.56, 120.64, 114.69, 54.44, 49.50, 44.78, 21.65. HRMS [ESI^+^]: calcd for C_20_H_22_ClN_5_O_2_S, 432.1255 [M + H]^+^; found, 432.1255.

4-(3,5-Dimethoxyphenyl)-*N*-(2-methoxythieno[2,3-*b*]pyrazin-3-yl)piperazine-1-carboxamide **7c**: ^1^H NMR (500 MHz, CDCl_3_) δ 7.49 (d, *J* = 5.9 Hz, 1H), 7.26 (d, *J* = 2.3 Hz, 1H), 7.13 (s, 1H), 6.11 (d, *J* = 1.8 Hz, 2H), 6.07 (s, 1H), 4.10 (s, 3H), 3.79 (s, 6H), 3.76–3.69 (m, 4H), 3.32–3.21 (m, 4H). ^13^C NMR (126 MHz, CDCl_3_) δ 161.56, 153.34, 152.89, 148.44, 144.94, 140.14, 136.55, 126.40, 121.42, 95.55, 92.19, 55.29, 54.26, 49.19, 44.72. HRMS [ESI^+^]: calcd for C_20_H_23_N_5_O_4_S, 430.1544 [M + H]^+^; found, 430.1544.

4-(3,5-Dimethylphenyl)-*N*-(2-methoxythieno[2,3-*b*]pyrazin-3-yl)piperazine-1 carboxamide **7d**: ^1^H NMR (500 MHz, CDCl_3_) δ 7.49 (d, *J* = 5.9 Hz, 1H), 7.29–7.23 (m, 1H), 7.12 (s, 1H), 6.58 (s, 3H), 4.11 (s, 3H), 3.77–3.69 (m, 4H), 3.29–3.20 (m, 4H), 2.30 (d, 6H). ^13^C NMR (126 MHz, CDCl_3_) δ 153.40, 148.54, 144.91, 140.21, 139.04, 136.61, 127.48, 126.40, 121.47, 121.31, 114.92, 54.28, 49.77, 44.68, 21.63 HRMS [ESI^+^]: calcd for C_20_H_23_N_5_O_2_S, 398.1645 [M + H]^+^; found, 398.1645.

4-(3,5-Difluorophenyl)-*N*-(2-methoxythieno[2,3-*b*]pyrazin-3-yl)piperazine-1-carboxamide **7e**: ^1^H NMR (500 MHz, CDCl_3_) δ 7.50 (d, *J* = 5.6 Hz, 1H), 7.26 (d, *J* = 4.5 Hz, 1H), 7.13 (s, 1H), 6.38 (d, *J* = 9.3 Hz, 2H), 6.31 (*t*, *J* = 8.6 Hz, 1H), 4.10 (s, 3H), 3.73 (s, 4H), 3.31 (s, 4H). ^13^C NMR (126 MHz, CDCl_3_) δ 164.03 (dd, J = 244.8, 15.8 Hz), 153.43 (s), 152.74 (*t*, J = 12.3 Hz), 148.45 (s), 144.86 (s), 140.27 (s), 136.44 (s), 126.51 (s), 121.49 (s), 98.57–98.28 (m), 94.79 (*t*, J = 26.1 Hz), 54.32 (s), 48.07 (s), 44.43 (s). HRMS [ESI^+^]: calcd for C_18_H_17_F_2_N_5_O_2_S, 406.1144 [M + H]^+^; found, 406.1144.

4-(3-methoxy-5-methylphenyl)-*N*-(2-methoxythieno[2,3-*b*]pyrazin-3-yl)piperazine-1-carboxamide **7f**: ^1^H NMR (500 MHz, CDCl_3_) δ 7.51 (d, *J* = 5.9 Hz, 1H), 7.29 (s, 1H), 7.15 (s, 1H), 6.41 (s, 1H), 6.33 (s, 2H), 4.12 (s, 3H), 3.79 (s, 3H), 3.78–3.66 (m, 4H), 3.36–3.22 (m, 4H), 2.34 (s, 3H). ^13^C NMR (126 MHz, CDCl_3_) δ 160.59, 153.33, 152.17, 148.45, 144.95, 140.13, 140.07, 136.57, 126.39, 121.41, 110.20, 106.28, 100.28, 55.19, 54.26, 49.28, 44.78, 22.02. HRMS [ESI^+^]: calcd for C_20_H_23_N_5_O_3_S, 414.1594 [M + H]^+^; found, 414.1594.

4-(3-Fluoro-5-methoxyphenyl)-*N*-(2-methoxythieno[2,3-*b*]pyrazin-3-yl)piperazine-1-carboxamide **7g**: ^1^H NMR (500 MHz, CDCl_3_) δ 7.50 (d, *J* = 5.9 Hz, 1H), 7.26 (d, *J* = 6.6 Hz, 1H), 7.14 (s, 1H), 6.24 (d, *J* = 13.4 Hz, 2H), 6.18 (d, *J* = 10.3 Hz, 1H), 4.10 (s, 3H), 3.78 (s, 3H), 3.73 (s, 4H), 3.28 (s, 4H). ^13^C NMR (126 MHz, CDCl_3_) δ 164.55 (d, J = 241.8 Hz), 161.60 (d, J = 13.5 Hz), 153.37 (s), 152.74 (d, J = 12.5 Hz), 148.44 (s), 144.91 (s), 140.19 (s), 136.50 (s), 126.44 (s), 121.44 (s), 98.08 (d, J = 2.4 Hz), 95.98 (d, J = 25.6 Hz), 92.81 (d, J = 25.9 Hz), 55.47 (s), 54.28 (s), 48.59 (s), 44.56 (s). HRMS [ESI^+^]: calcd for C_19_H_20_FN_5_O_3_S, 418.1344 [M + H]^+^; found, 418.1344.

4-(3-Fluoro-5-methylphenyl)-*N*-(2-methoxythieno[2,3-*b*]pyrazin-3-yl)piperazine-1-carboxamide **7h**: ^1^H NMR (500 MHz, CDCl_3_) δ 7.50 (d, *J* = 5.8 Hz, 1H), 7.26 (d, *J* = 5.6 Hz, 1H), 7.13 (s, 1H), 6.51 (s, 1H), 6.42 (d, *J* = 10.2 Hz, 2H), 4.10 (s, 3H), 3.73 (s, 4H), 3.27 (s, 4H), 2.31 (s, 3H). ^13^C NMR (126 MHz, CDCl_3_) δ 163.76 (d, J = 243.1 Hz), 153.37 (s), 152.24 (d, J = 10.4 Hz), 148.44 (s), 144.92 (s), 140.78 (d, J = 9.7 Hz), 140.18 (s), 136.52 (s), 126.43 (s), 121.44 (s), 112.42 (d, J = 2.1 Hz), 107.56 (d, J = 21.3 Hz), 100.45 (d, J = 25.1 Hz), 54.28 (s), 48.81 (s), 44.64 (s), 21.81 (s). HRMS [ESI^+^]: calcd for C_19_H_20_FN_5_O_2_S, 402.1395 [M + H]^+^; found, 402.1395.

4-(3,5-dimethoxyphenyl)-*N*-(2-ethoxythieno[2,3-*b*]pyrazin-3-yl)piperazine-1-carboxamide **7i**: ^1^H NMR (500 MHz, CDCl_3_) δ 7.47 (d, *J* = 5.9 Hz, 1H), 7.23 (d, *J* = 5.9 Hz, 1H), 7.13 (s, 1H), 6.11 (d, *J* = 1.8 Hz, 2H), 6.07 (s, 1H), 4.53 (q, *J* = 7.1 Hz, 2H), 3.79 (s, 6H), 3.76–3.68 (m, 4H), 3.32–3.21 (m, 4H), 1.48 (*t*, *J* = 7.1 Hz, 3H). ^13^C NMR (126 MHz, CDCl_3_) δ 161.56, 153.47, 152.91, 147.89, 144.65, 139.99, 136.59, 126.09, 121.48, 95.55, 92.19, 63.10, 55.29, 49.20, 44.84, 14.48. HRMS [ESI^+^]: calcd for C_21_H_25_N_5_O_4_S, 444.1700 [M + H]^+^; found, 444.1700.

### 3.9. General Antibodies and Reagents

The following antibodies were used: c-Jun (#9156, Cell signaling, Beverly, MA, USA, 1:1000); p-c-Jun (sc-822, Santa Cruz Biotechnology, Dallas, TX, USA, 1:500); cyclin B1 (sc-7393, Santa Cruz Biotechnology, Dallas, TX, USA, 1:500); CDC4 (sc-331296, Santa Cruz Biotechnology, Dallas, TX, USA, 1:250); Ubiquitin (#3936, Cell signaling, Beverly, MA, USA, 1:1000); β-actin (sc-47778, Santa Cruz Biotechnology, Dallas, TX, USA, 1:500). pCMV-Myc-tagged CDC4 wild type plasmid (#16652) was purchased from Addgene (Watertown, MA, USA). Construction of the Myc tagged-CDC4 double-point mutant-type plasmid (R689G, T628A) was based on wild-type plasmids.

### 3.10. Cell Culture and siRNA

The HCT116, HCT8, KM12, HCT15, HT29, COLO-205, SW620 cell line was obtained from the National Cancer Institute (Material Transfer Agreement number: 2702–09). DLD1 cell line was obtained from the Korean cell line bank. Cells were cultured in complete Roswell Park Memorial Institute (RPMI) 1640 medium (Hyclone, South Logan, UT, USA) containing 10% fetal bovine serum (Hyclone, South Logan, UT, USA) in an atmosphere of 5% CO_2_ and 100% humidity at 37 °C. A small interfering RNA (siRNA) duplex targeting human c-Jun and FBXW7 (Genolution, Seoul, Korea) were introduced into cells using Lipofector-EXT (AptaBio, Yongin, Korea), according to the manufacturer’s instructions. As negative controls, cells were incubated with Lipofector-EXT (AptaBio, Yongin, KOR) and a negative siRNA (Genolution, Seoul, Korea).

### 3.11. Sulforhodamine B (SRB) Assay

Cells (100 μL containing 5000–10,000 cells/well) were incubated in 96 well microtiter plates. After 24 h, drugs were added (100 μL) to each well and the cultures were incubated for 48 h at 37 °C. The cells were then fixed in TCA (50 μL per well). The plates were incubated for a minimum of 1 h or a maximum of 3 h at 4 °C. The liquid was removed from the plate, which was then rinsed five times with water and allowed to dry at room temperature (rt) for approximately 12–24 h. The fixed cells were stained with 100 μL Sulforhodamine B (SRB) solution for 5 min at rt. After staining, the plate was washed three times with 1% glacial acetic acid and dried at rt for approximately 12–24 h. The SRB was then solubilized in 10 mM Tris buffer, and the absorbance was read at 515 nm. The effect of the drugs was expressed as GI50 (50% growth inhibition).

### 3.12. Western Blotting

The whole-cell lysate was prepared using Radioimmunoprecipitation assay buffer (RIPA buffer) prepared in 50 mM Tris-HCl, pH 8.0, with 150 mM sodium chloride, 1.0% Igepal CA-630 (NP-40), 0.5% sodium deoxycholate, 0.1% sodium dodecyl sulfate, protease inhibitor cocktail, and phosphatase inhibitor cocktail. Protein assays were carried out to normalize the proteins using a Bradford protein assay (Thermo Scientific, Waltham, MA, USA). Proteins were resolved by SDS-PAGE and were transferred to polyvinylidene difluoride (PVDF) membrane (Merck Millipore, Burlington, MA, USA). Membranes were blocked in 5% bovine serum albumin (BSA) for 1 h at rt and incubated with indicated antibodies overnight at 4 °C. Membranes were washed in Tris-buffered saline, 0.1% Tween 20 (TBST) for 1 h at rt and incubated with horseradish peroxidase-conjugated secondary antibody for 1 h at rt. Membranes were washed five times with TBST, and chemiluminescence was detected using Westsave™ (Abfrontier, Seoul, Korea). Gels were imaged using FUSION-Solo.4.WL (Vilber Lourmat, Collégien, France).

### 3.13. Immunoprecipitation

To identified ubiquitination of c-Jun by DGG200064, HCT116 cells were treated with/without MG132 (10 μM) for 6h, DGG200064 (50 nM) for 18 h. The cell lysate was prepared using RIPA buffer for immunoprecipitation. Each lysate was mixed with antibodies (1 μg/mL) of c-Jun or MYC at 4 °C overnight in immunoprecipitation buffer containing 50 mM Tris-Cl,150 mM NaCl, 1 mM EDTA, 0.5% Triton X-100, pH 7.4, and reacted with 10 μL of protein A/G beads UltraLink Resin (50:50 resin: buffer slurry condition) (Peirce, #35133) for 2 h on RT for precipitation. After reaction and centrifugation at 3000 rpm for 3 min, the immunoprecipitated samples were washed with 500 μL of immunoprecipitation buffer by tapping and centrifuged. The washing was repeated five times before western blotting.

### 3.14. Preclinical Xenograft Tumor Models

Xenografts were initiated in female Bagg and Albino (BALB)/c nude mice (6–8 weeks old; *n* = 18). Briefly, HCT116 cells (5.0 × 106) were inoculated subcutaneously. After 1 week, the mice were divided into three groups of six mice each, a control group treated with vehicle only (10% Dimethyl sulfoxide (DMSO); 50% Polyethylene glycol (PEG400); 40% Phosphate-buffered saline (PBS)) and inhibitor (DGG200064 and DGG200338)—treated two groups. After a 2-week evaluation of the maximum tolerated dose of DGG200064 and DGG200338, we observed no lethality and no loss of body weight with a dose of 150 mg/kg by oral administration. Therefore, we decided to use 60 mg/kg, 120 mg/kg. Vehicle alone and inhibitor (DGG200064 60 mg/kg, 120 mg/kg) (DGG200338 35 mg/kg, 70 mg/kg) were administered orally once per day, 6 days/week, for 21 days. The size of the primary tumors was measured every 2–3 days using calipers. Tumor volume was calculated using the formula, *v* = (A × B2)/2, where V is the short diameter (mm). Mice were injected intraperitoneally with 10 mg/mL Hematoxylin in sterile Dulbecco’s PBS (DPBS). After 2 h, the animals were anesthetized perfused with PBS and killed, followed by excision of the tumors. The study was reviewed and approved by the Institutional Animal Care and Use Committee (IACUC) of the National Cancer Center Research Institute. The National Cancer Center Research Institute (NCCRI) is an Association for Assessment and Accreditation of Laboratory Animal Care International (AAALAC International)-accredited facility and abides by the Institute of Laboratory Animal Resources (ILAR) guidelines (IRB number: NCC-15–245, approval date. 26.03.15).

### 3.15. Immunohistochemistry (IHC)

The isolated tumors issued were then stained with antibodies against BrdU (1:700), c-Jun (1:50). The staining solutions used for IHC were obtained from Thermo Scientific (Waltham, MA, USA). All procedures were performed on 4 μm sections of paraffin-embedded tissue at rt. Antigen retrieval was performed on formalin-fixed tissue in a pressure cooker at 121 °C for 20 min and was followed by IHC. Endogenous peroxidase activity was blocked with hydrogen peroxide, and background staining was blocked with V block solution, the primary antibodies were detected with horseradish peroxide (HRP)-conjugated polymer and developed with diaminobenzidine (DAB). The sections were then counterstained with hematoxylin, dehydrated in graded series of alcohols, and mounted with Richard-Allan Scientific mMounting Media from Thermo Scientific (Waltham, MA, USA). Representative images from each tumor were collected using a 20× objective lens. Positively stained nuclei were counted, and the differences between the groups were analyzed using Student’s *t*-test, *p* values ≤0.05 were considered significant.

### 3.16. Cell Cycle Distribution

A colon cancer cells were trypsinized and fixed in 70% (vol/vol) ethanol for DNA staining. For cell cycle analysis, fixed cells were washed with PBS and subsequently resuspended in Propidium Iodide (PI)/RNase staining solution (0.05 mg/mL PI, 0.1 mg/mL RNase A in PBS). FACS analysis was performed using FACSCanto II (BD Biosciences, San Jose, CA, USA).

### 3.17. Immunofluorescence Analysis

HCT116 cells were grown on coverslips and fixed in 4% paraformaldehyde for 10 min at RT. Fixed cells were stained with anti-c-Jun antibodies. Alexa Fluor 546 (red) conjugated secondary antibodies were used for visualization. DAPI was used to stain the nuclei (blue). Cells incubated with secondary antibodies alone were used as controls. Images were obtained using a Zeiss Axiovert 200M microscope.

### 3.18. Statistical Analysis

Statistical analysis was performed using Student’s *t*-test. Tumor growth in the xenograft mouse model was analyzed statistically by two-way analysis of variance (ANOVA) using Microsoft Excel. A *p*-value less than 0.05 (typically ≤ 0.05) is statistically significant.

### 3.19. Pharmacokinetic and Physicochemical Analysis

Sprague–Dawley rats (adult males, 250–300 g) were fasted overnight, and the femoral vein (for iv administration of the compound) and jugular vein (for blood sampling) of each rat were cannulated with polyethylene tubing. The selected compounds dissolved in 30% polyethylene glycol (PEG) in saline were administered either intravenously or orally. At various time points after administration, blood samples (0.5 mL) were collected from the jugular vein, transferred to heparin-coated tubes, centrifuged to separate off the plasma. The plasma was stored at −80 °C until assayed. Compound concentrations in plasma were determined by LC/MS/MS analysis (API4000 mass spectrometer, Applied Biosystem, Waltham, MA, USA). Pharmacokinetic parameters were analyzed using the WinNonlin software program. The area under the curve (AUC) was calculated using the trapezoidal rule extrapolated to infinity. The terminal elimination half-life, systemic clearance, and volume of distribution at steady state were determined. The extent of absolute oral bioavailability (F) was estimated by comparing the AUC values after intravenous and oral administration of the compounds.

## 4. Conclusions

In this study, we have designed and synthesized a novel of thieno[2,3-*b*]pyrazine derivatives with various substitutions on both thienopyrazine and phenyl rings and tested them against human colorectal cancer cell lines to identify the final lead compound. We selected compound **DGG200064** (c), with the 3,5-dimethoxy group on the phenyl ring and another methoxy group on the thieno[2,3-*b*]pyrazine core, as the lead compound.

We demonstrated growth inhibition of colorectal cancer through the G2/M phase arrest by **DGG200064**. Our results confirmed a dose-dependent increase of c-Jun, phosphorylated c-Jun, and cyclin B1 with **DGG200064** treatment. We indicated that the increase in the G2/M phase arrest marker, cyclin B1, is regulated by c-Jun; and c-Jun protein degradation is regulated by an E3 ligase FBXW7. Moreover, our results confirmed that **DGG200064** selectively inhibited c-Jun ubiquitination by FBXW7 in colon cancer cell lines: **DGG200064** binds to FBXW7 (aa 626–689) and inhibits its interaction with c-Jun.

In this study, several properties of the representative compounds were analyzed: in vitro and in vivo cell line tests including resistance cell lines, pharmacokinetic study, calculation of the physicochemical properties, and various toxicity assays. The results demonstrated that **DGG200064** has strong novel anticancer efficacy and desirable orally druggable properties.

## Data Availability

Data is contained within the article and Appendix A.

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
