# Peer review of "Design and Synthesis of a Novel 4-aryl-N-(2-alkoxythieno [2,3-b]pyrazine-3-yl)-4-arylpiperazine-1-carboxamide DGG200064 Showed Therapeutic Effect on Colon Cancer through G2/M Arrest"

_pharmaceuticals, 2022, doi:10.3390/ph15050502_

Round 1
Reviewer 1 Report
Comments to Author:
- I suggest to the authors add figure on the first page, helping the readers to understand the background of this work.
- In scheme 1 the 5c and 5d compounds were prepared by using 5a & 5b compounds. So, make the arrow form the 5a,b to 5c,d. Readers can understand more easily.
- In scheme 2, the authors mentioned both R2 and phenyl formate on -N. Please change it by deleting the phenyl formate.
- On line 95 the sentence is like “the dichlorination of the compounds 5a-5b” instead of ‘dichlorination’. Change it.
- The authors claimed that the two compounds 7c and 7f have good cytotoxicity and SAR analysis. But the authors further studied only the 7c compound, did they check the activity of the 7f compound?
- The figure 1b graph letters are not visible clearly, please make sure that.
- In figure 4b, the structure needs to be correct to -OMe not -OEt.
- On line 53 it should be “which can”.
- I suggest adding the Rf value and ratio of solvents used for the chromatography purification of each compound.
Author Response
Please find an attached file.

Reviewer 2 Report
In the present manuscript, titled "Design and synthesis of a novel 4-aryl-N-(2-alkoxythieno[2,3-b]pyrazine-3-yl)-4-arylpiperazine-1-carboxamide DGG200064 showed therapeutic effect on colon cancer through G2/M arrest", the authors describe the synthesis, mechanism of action, efficacy, pharmacokinetic and physicochemical analysis of the compound DGG200064 in colon cancer model. The manuscript is well drafted and tries to deliver evidence in the favor of development of DGG200064 for colon cancer. There are some concerns regarding the paper:
Major concerns:
- Lines 75-77: Has the compound DGG200064 or similar compounds been studied before as referred to in these lines (reference 1b)? If yes, please elaborate on the novelty of using this compound in the introduction compared to previous works.
- The levels of FBXW7 after siRNA treatment change slightly in HCT116 cells (fig 2d). Loading control b-actin is also quite variable. Please quantify all western blot densities using ImageJ and put them together with the figures in the results.
- What were the in vivo biochemical effects of the compound DGG200064? Evaluate cardiac troponin I levels and analysis of biochemical analytes (renal tests, liver, and muscle enzyme activity).
Minor concerns:
- Line 42: replace "Pan-CDKs" with "Pan-CDK"
- Line 53: edit "can"
- line 54: remove "studies"
- line 77: replace "CDKs" with "CDK"
- line 179: remove "The"
- Please re-check for additional grammatical/syntactical errors.
Author Response
Please find an attached file.

Reviewer 3 Report
The manuscript presented the well-planned analysis of the new lead compound, with a 2-alkoxythieno[2,3-b]pyrazine-3-yl)-4-arylpiperazine-1-carboxamide core skeleton as the anticancer drug in colon cancer development. It is especially crucial because colon cancer is still characterized by high mortality.
Despite that, I found some significant problems in the experiments and their interpretation:
1. How do the authors interpret the different G2/M cell cycle arrest times in particular colon cancer cells (Fig 1).
2. If each line reaches G2/M cell cycle arrest, the authors should have performed the analysis on all lines in fig 2 and fig 3
3 It is incomprehensible to select only one line for animal analysis.
A minor note:
1 The description of biological methods should be included in the main manuscript
Author Response
Please find an attached file.

Round 2
Reviewer 2 Report
No further questions.
Author Response
No further questions
Reviewer 3 Report
Ad 1 The authors did not answer my first question.
They corrected their mistake but did not interpret the result logically and correctly.
Ad 2 The answer to the second question is also incorrect. I am not interested in the conditions of the experiment but in the fact that the analyses should be performed on all cell lines. I do not understand, and the authors do not explain why some lines were omitted. They should add the reasons for the limitations.
Ad 3 Even if the studies are carried out only as proof of concept, they should also be carried out on the DLD1 line. The assumption of the animal studies, as well as their results from the only one cell lines, do not fulfill even such a limited goal.
Author Response
Ad 1 The authors did not answer my first question.
They corrected their mistake but did not interpret the result logically and correctly.
Ans) We are sorry for the wrong answer. The cell cycle and cell division depend on the function of the cancer cell and the type of mutation. Stem cell like cancer cell showed slow cell cycle and differentiated cancer cell showed fast cell cycle. Therefore, G2/M arrest may vary among cancer cells. Therefore, G2/M arrest may vary between cancer cells. In this study, we observed cell death through a microscope 24 hours after treatment with DGG20064. Our results showed G2/M cell cycle arrest under conditions when colon cancer cell lines did not die after 6 h of drug treatment.
In the case of nocodazole treatment for G2/M arrest, it has been reported that G2/M arrest occurs from 6 hours to 24 hours after drug treatment in breast cancer cell lines.
Ref) Hye J Choi.; Masayuki F.; Bao T Zhu. Role of Cyclin B1/Cdc2 Up-Regulation in the Development of Mitotic Prometaphase Arrest in Human Breast Cancer Cells Treated with Nocodazole. PLoS One. 2011; 6(8): e24312.
Ad 2 The answer to the second question is also incorrect. I am not interested in the conditions of the experiment but in the fact that the analyses should be performed on all cell lines. I do not understand, and the authors do not explain why some lines were omitted. They should add the reasons for the limitations.
Ans) We showed that the drug induced G2/M arrest in a concentration-dependent manner for all cell lines in Fig 1. Fig 2 showed representative data of increase of the expression of c-JUN or cyclin B1 as the G2/M arrest marker. In molecular biology experiments, usually two different cell lines are required to show proof of concept.
Ad 3 Even if the studies are carried out only as proof of concept, they should also be carried out on the DLD1 line. The assumption of the animal studies, as well as their results from the only one cell lines, do not fulfill even such a limited goal.
Ans) Comments are under investigation. We are investigating anti-cancer effect of the compound against other colon cancer cell lines using xenograft model. However, it takes time and efforts for very long period (at least a half year). Here, we present three groups of drug dosages which showed dose dependent anti-cancer effect. Here we demonstrated proof of concept only. Detailed application by cancer cell types with differential mutations will be investigating in near future. Thank you.
Round 3
Reviewer 3 Report
Ad 1 and 2) I do not know why the authors assumed that analysis of only two different cell lines is needed to prove a proof of concept. Such a priori assumption is not an appropriate research approach. Even if the authors would like to limit the study to such a small number of cell lines, they should give the reason for their choice or choose the extreme-behaving lines. Unfortunately, we have not observed that effect (Figure 1), so I am completely confused by their decision and a complete lack of respect for the suggestions of the reviewer trying to lead the authors to proper experimental approach.
To consider the result presented in figure 2, the authors have to give a cell lines choice based on a logical analysis of the results presented earlier.
Ad 3) The results presented on only one cell line in animal model are unreliable. Because the authors again did not provide the reason for their choice I not understand their decision. It the present version, the manuscript should be revised and submitted in the rewritten version after completing this lack part of the study. Otherwise, it does not meet the publication requirements.